# Mouse Acidic Chitinase Effectively Degrades Random-Type Chitosan to Chitooligosaccharides of Variable Lengths under Stomach and Lung Tissue pH Conditions

**DOI:** 10.3390/molecules26216706

**Published:** 2021-11-05

**Authors:** Satoshi Wakita, Yasusato Sugahara, Masayuki Nakamura, Syunsuke Kobayashi, Kazuhisa Matsuda, Chinatsu Takasaki, Masahiro Kimura, Yuta Kida, Maiko Uehara, Eri Tabata, Koji Hiraoka, Shiro Seki, Vaclav Matoska, Peter O. Bauer, Fumitaka Oyama

**Affiliations:** 1Department of Chemistry and Life Science, Kogakuin University, Tokyo 192-0015, Japan; wsat.chorus.618@gmail.com (S.W.); bt79310@ns.kogakuin.ac.jp (Y.S.); nakamura.shibatagousei@gmail.com (M.N.); shuukatsu1215@gmail.com (S.K.); heimonana@icloud.com (K.M.); bm19024@g.kogakuin.jp (C.T.); bd17002@g.kogakuin.jp (M.K.); yuta.maldini@blue.email.ne.jp (Y.K.); bd19001@ns.kogakuin.ac.jp (M.U.); bq21002@ns.kogakuin.ac.jp (E.T.); 2Japan Society for the Promotion of Science (PD), Tokyo 102-0083, Japan; 3Department of Environmental Chemistry, Kogakuin University, Tokyo 192-0015, Japan; bm18040@g.kogakuin.jp (K.H.); shiro-seki@cc.kogakuin.ac.jp (S.S.); 4Laboratory of Molecular Diagnostics, Department of Clinical Biochemistry, Hematology and Immunology, Homolka Hospital, Roentgenova 37/2, 150 00 Prague, Czech Republic; vaclav.matoska@homolka.cz (V.M.); peter.bauer@bioinova.cz (P.O.B.); 5Bioinova JSC, Videnska 1083, 142 20 Prague, Czech Republic

**Keywords:** acidic chitinase, block-type chitosan, chitin, chitooligosaccharides, FACE method, random-type chitosan

## Abstract

Chitooligosaccharides exhibit several biomedical activities, such as inflammation and tumorigenesis reduction in mammals. The mechanism of the chitooligosaccharides’ formation in vivo has been, however, poorly understood. Here we report that mouse acidic chitinase (Chia), which is widely expressed in mouse tissues, can produce chitooligosaccharides from deacetylated chitin (chitosan) at pH levels corresponding to stomach and lung tissues. Chia degraded chitin to produce *N*-acetyl-d-glucosamine (GlcNAc) dimers. The block-type chitosan (heterogenous deacetylation) is soluble at pH 2.0 (optimal condition for mouse Chia) and was degraded into chitooligosaccharides with various sizes ranging from di- to nonamers. The random-type chitosan (homogenous deacetylation) is soluble in water that enables us to examine its degradation at pH 2.0, 5.0, and 7.0. Incubation of these substrates with Chia resulted in the more efficient production of chitooligosaccharides with more variable sizes was from random-type chitosan than from the block-type form of the molecule. The data presented here indicate that Chia digests chitosan acquired by homogenous deacetylation of chitin in vitro and in vivo. The degradation products may then influence different physiological or pathological processes. Our results also suggest that bioactive chitooligosaccharides can be obtained conveniently using homogenously deacetylated chitosan and Chia for various biomedical applications.

## 1. Introduction

Chitin is a polysaccharide abundantly occurring in nature and consisting of β-1,4-linked *N*-acetyl-d-glucosamine (GlcNAc). It functions as the major structural component of the exoskeleton of crustaceans and insects, microfilarial shells of nematodes, and cell walls of fungi [1]. Chitin exists in three major forms—α [2,3,4], β [4], and γ [5,6]—differing in the orientation and packing of the polymeric chains.

Chitosans form a large family of heteropolymers of d-glucosamine (GlcN) and GlcNAc residues at various ratios and patterns. They are partially deacetylated derivatives of chitin.

Chitosan can be prepared from chitin by two fundamentally different methods—heterogeneous and homogeneous deacetylation [7]. Heterogeneously deacetylated chitosan (hereafter called “block-type chitosan”) with a deacetylation degree (DD) of up to 80% maintain certain degree of crystallinity. Homogeneous deacetylation results in random-type polymers (hereafter called “random-type chitosan”) [8]. The random-type chitosan with a DD of ~50% is soluble in water. The crystallinity of the molecules decreases rapidly with the DD of >40% and these molecules are defined as amorphous [8].

Although humans and mice do not synthesize endogenous chitin, they produce two active chitinases, chitotriosidase, and acidic chitinase (hereafter referred to as “Chia”; also reported as acidic mammalian chitinase, “AMCase”) [9,10,11,12,13]. Chia has been an attractive subject in biomedical research because its levels are altered in various diseases such as asthma, allergic inflammation, gastric cancer, ocular allergy, and dry eye syndrome [14,15,16,17,18,19,20]. In addition, polymorphisms in the Chia gene are associated with bronchial asthma in humans [21,22,23]. Recently, it has also been shown that chitinase activity in mouse airways depends on Chia [24,25].

Mouse Chia mRNA is predominantly expressed in the stomach [13,26,27], followed by the submaxillary gland and lung [26]. Recombinant Chia has peak activity at a pH of around 2.0 that gradually decreases at more neutral conditions (pH 3.0–7.0), primarily producing (GlcNAc)_2_ [28,29,30,31]. In addition, mouse Chia exhibits transglycosylation activity under neutral conditions [32]. Chia is resistant to digestion by proteases in mice, chickens, pigs, and common marmosets under gastrointestinal-like conditions [33,34,35,36,37].

Several recent reports have indicated that chitooligosaccharides derived from chitin and chitosan are biologically active and have anti-tumorigenic and anti-inflammatory effects [38,39,40,41]. Chitooligosaccharides can be produced by chemical or enzymatic methods [42,43]. Enzymatic preparation by nonspecific enzymes—such as cellulases, lipases, chitosanases, and proteases, including commercially available porcine pepsin preparations—have been used to produce chitooligosaccharides with various biomedical activities [44,45,46].

We have recently reported that Chia residues in porcine pepsin preparations exhibit chitinolytic activity under the stomach condition [47]. It remains elusive whether Chia can efficiently digest chitosan under physiological conditions of peripheral tissues, such as the lungs.

In this study, we show that mouse Chia produces chitooligosaccharides under such conditions and that the random-type chitosan is more suitable for efficient and production of oligomers of variable sizes when compared with the block-type form of the molecule [7,8].

## 2. Results

### 2.1. Degradation of α- and β-Colloidal Chitin Substrates

We incubated α- or β-colloidal chitin (α- or β-chitin) at pH 2.0 and 37 °C for 1, 24, or 72 h and analyzed the degradation products by the fluorophore-assisted carbohydrate electrophoresis (FACE) method.

We mainly detected (GlcNAc)_2_ from α-chitin, whereas β-chitin was degraded into chitooligosaccharides of different sizes—including dimers, trimers and, with lower efficiency, also longer molecules (Figure 1). As we have reported previously, element analysis showed a higher degree of deacetylation (DD) in β-chitin when compared to that of α-chitin [47]. Thus, the relationship between higher chitin DD and the production of longer chitooligosaccharides observed with porcine Chia [47] can also be seen in case of mouse Chia.

### 2.2. Characterization of Block-Type and Random-Type Chitosan Substrates by X-ray Diffraction

We analyzed the relationship between the degradation pattern, deacetylation mode and DD. We characterized the block- and random-type chitosan as well as α- and β-chitin substrates by X-ray diffraction, as described in Materials and Methods. α- and β-chitin exhibited characteristic peaks of 2θ = 9.5° and 19.5°, and 2θ = 8.36° and 19.72°, respectively (Figure 2), consistently with previous reports [6,8,48].

Block-type chitosans with 69%, 73%, 84%, and 95% DD (chitosan 7B, 8B, 9B, 10B, Funakoshi Co., Ltd., Tokyo Japan) showed characteristic pattern (2θ = 20.06 to 20.30°) responding to DD and peaking at 84% DD (Figure 2) that represents the amorphous form of the molecule. Random-type chitosan shows the pattern of an amorphous molecule [8].

### 2.3. Degradation of Block-Type Chitosan

Next, we examined the degradation of block-type chitosans (Table 1). This type of chitosan is soluble at pH 2.0, which is the optimal condition for Chia [13,28]. The substrates were dissolved in McIlvaine buffer (pH 2.0), incubated with Chia at 37 °C for 1, 24, or 72 h and analyzed by the FACE method.

Chia digested the substrates with up to 84% DD primarily to (GlcNAc)_2_, (GlcNAc)_3_, (GlcNAc)_6_, and (GlcNAc)_9_ (Figure 3). More prolonged incubation (24 h and 72 h vs. 1 h) markedly increased the yield of the degradation. With increasing DD, the reaction efficiency decreased, and 95% DD chitosan remained undigested (Figure 3). Thus, mouse Chia acts in this regard in a similar manner as the porcine enzyme [47]. These results indicate that mouse Chia has primarily chitinase but not chitosanase activity.

### 2.4. Degradation of Random-Type Chitosan

We next aimed to analyze how Chia degrades the random-type chitosan at different pH conditions. In contrast to the block-type chitosan, this type of chitosan is directly soluble in water (Table 1). Thus, the solubilized substrates with 36% and 45% DD were incubated with Chia in the McIlvaine buffer at pH 2.0, 5.0, or 7.0 for 1 h and analyzed by the FACE method. Figure 4A shows chitooligosaccharides with various sizes generated from the 36% DD chitosan under-tested pH conditions. Prolonged incubation (24 and 72 h) markedly increased the yield but did not change the ratio of the chitooligosaccharides.

Processing of the 45% DD molecule provided similar results (Figure 4B). However, we observed pH dependency with higher and lower DD leading to increased degradation products at pH 2.0 and 7.0, respectively, at 24 h. Thus, the DD of the random-type chitosan can influence the degradation of the substrates by Chia under different pH conditions.

### 2.5. Chia Produces Chitooligosaccharides more Efficiently and with Higher Size Variability from Random-Type Chitosan

We compared the degradation products of block-type chitosan with 69% and random-type chitosan with 45% DD. The substrates were dissolved directly in McIlvaine buffer (pH 2.0), incubated with Chia at 37 °C for 1, 24, or 72 h and analyzed by the FACE method. The pattern and efficiency of the degradation differed significantly between the two types of chitosan substrates (Figure 5, upper). Furthermore, more chitooligosaccharides can be obtained by prolonged incubation with Chia (24 and 72 h) than by the 1-h incubation (Figure 5, upper). Quantitative analysis indicated that higher levels of chitooligosaccharides with more variable lengths were produced from random-type chitosan rather than from the block-type form of the molecule (Figure 5, lower). In the random-type chitosan, the distribution of the chitooligosaccharides was more even, but the variability in terms of the oligomers’ presence was very similar. These results suggest that the differences in the degradation efficiency and pattern can be affected by the presence of crystal structure and the DD mode of the substrates. It appears that the DD can change the substrate affinity to the enzyme.

## 3. Discussion

We have previously shown that mouse Chia degrades α-chitin exclusively to (GlcNAc)_2_ [28,30,31]. Here we show that mouse Chia produces chitooligosaccharides from the block- and random-type chitosan substrates under pH conditions present in different mammalian tissues.

Chitosan and chitooligosaccharides have attracted substantial interest due to their biomedical activities that include anti-microbial [49], anti-inflammatory [50], and anti-tumor effects [51]. It remains, however, unclear whether chitooligosaccharides are spontaneously produced in vivo. We showed that Chia could degrade block-type chitosan substrates at pH 2.0 and random-type chitosan at pH 2.0–7.0. The quantitative FACE analysis revealed that the degradation product pattern of chitooligosaccharides differs depending on the substrate and reaction conditions. The produced chitooligosaccharides may trigger several cascades, such as anti-pathogen and immune responses in the living organisms. These data can be applied to in vivo studies aiming to understand the physiological importance of the chitooligosaccharides.

More efficient degradation with higher chitooligosaccharides’ size variability was observed from the random-type chitosan as compared to the block-type chitosan. Our results suggest that structural differences between chitin and chitosan can regulate the substrate specificity of Chia and can significantly alter the composition of the resulting degradation products. The block-type chitosan has amorphous regions, which are highly deacetylated areas (GlcN-rich), so Chia targets a cluster of the acetylated areas and produces mainly dimers and trimers (Figure 6, left). In contrast, when targeting the random-type chitosan, Chia preferentially degrades the randomly placed GlcNAc-rich regions to produce more variable-sized chitooligosaccharides (Figure 6, right). Random-type chitosan with DD around 50% is water-soluble and more versatile than other chitosan substrates. Thus, we propose the random-type chitosan as a suitable substrate for producing chitooligosaccharides in terms of quantity and quality for agricultural, pharmaceutical, and medical purposes.

Interestingly, DD of the random-type chitosan affected the pH-dependence of the degradation rate (Figure 4A,B). We have previously reported that mouse Chia shows higher transglycosylation and weaker glycosidase activity under neutral conditions than at acidic pH [32]. Importantly, substrate degradation and transglycosylation by Chia can occur simultaneously. Since Chia does not recognize the deacetylated (GlcN) regions, the enzyme degrades the GlcNAc areas formed through Chia’s transglycosylation of the substrate. Thus, 45% DD chitosan was degraded more efficiently than the substrate with 35% DD. We speculate that the reaction mode and efficiency generally depends on the combination of DD and pH.

The degradation products in this study were evaluated using the FACE method. This method is useful in analyzing sugars with various reducing ends, including chitooligosaccharides [30,52]. When compared with HPLC, which is often used to analyze chitooligosaccharides, there is a correlation between the two analytical methods in sugar identification and the degree of polymerization. Furthermore, FACE has several advantages: simple handling, high sensitivity, and low experimental costs [30]. However, this method is not able to analyze sugars that form the chitooligosaccharides [47]. To analyze the structure of individual degradation products, combination with NMR and MS analysis is necessary.

In agriculture, insect-based diets improve the growth performance and nutrient digestibility without affecting immune responses in poultries and pigs [53,54]. Furthermore, supplementation of chitooligosaccharides can enhance the immune response and function as an antibiotic/probiotic in pregnant pigs [54]. Recently, we have detected chitooligosaccharides after mealworm larvae or fly wing processing under chicken, pig, and marmoset gastrointestinal-like conditions [34,35,37]. The product patterns were similar to those obtained by chitosan digestion [34]. Our results showing Chia selectively degrading acetylated (chitin-like) areas of chitosan suggest that chitooligosaccharides can be produced in mammalian organisms from partially deacetylated chitin-containing organisms.

Since Chia expression is significantly altered under several pathological conditions such as asthma and allergic inflammation [14,15], it has attracted substantial scientific attention. Recently, Chia was shown to be a constitutively produced enzyme essential for chitin degradation in bronchial airways to maintain lung functions [25,55]. Our present results indicate that Chia can degrade various types of chitosan under acidic to neutral conditions. They suggest that Chia can assist in the removal of invaded chitin-containing pathogens, such as mites and molds, while producing chitooligosaccharides that may have in addition local anti-inflammatory effects.

Recently, we reported that crab-eating monkey Chia has robust chitinolytic activity under broad pH and temperature ranges [56]. This enzyme is 2×, 16×, and 10× more active at pH 2.0, 5.0, and 7.0, respectively, than mouse Chia. We suggest monkey Chia as a useful tool to act under various conditions to produce bioactive chitooligosaccharides efficiently. Further scrutiny on the chitosan degradation by monkey Chia is required to confirm its ability.

## 4. Materials and Methods

### 4.1. Preparation of Recombinant Mouse Chia and Measurement of Chitinase Activity

We expressed mouse Chia as a fusion protein of Protein A-Chia-V5-His in *Escherichia coli*, followed by its purification as described previously [28,29,57]. The chitinolytic activity was determined using 4-nitrophenyl *N*,*N*′-diacetyl-β-d-chitobioside [4-NP-(GlcNAc)_2_, Sigma-Aldrich, St. Louis, MO, USA] as reported previously [28,29,30]. Chia unit definition has also been described previously [28].

### 4.2. Preparation of α- or β-Colloidal Chitin

Chitin samples were powdered in a Wiley mill (Thomas Scientific, Swedesboro, NJ, USA) to particles of approximately 250 μm. We prepared α- or β-colloidal chitin from shrimp shells (Sigma-Aldrich) or squid pen chitin (Katakura & Co-op Agri Corporation, Tokyo, Japan) as described previously [28]. The suspended colloidal chitin was collected as a supernatant (10 mg/mL). α- or β-Colloidal chitin (2 mg/mL) was incubated with Chia (50 μU) in 50 μL of McIlvaine buffer (pH 2.0) at 37 °C for 1, 24, or 72 h.

### 4.3. Deacetylated Chitosan Substrates

The block-type chitosan was a generous gift from Funakoshi Co., Ltd. (Tokyo, Japan) and was described previously [47]. The random-type chitosan substrates were prepared from α-chitin, essentially described by Kurita et al. [8]. We determined the degree of deacetylation (DD) by elemental analysis as described previously (Table 1) [42].

### 4.4. X-ray Diffraction

Chitin and chitosan samples were powdered in the Wiley mill to a particle size of approximately 74 μm. The phase compositions of chitin and chitosan samples were recorded at room temperature by X-ray diffraction (XRD) patterns (Rigaku Miniflex 600, Tokyo, Japan) using Cu-Kα radiation (λ = 1.5406), in the 2θ range from 5° to 40°.

### 4.5. Degradation of Block-Type Chitosan by Chia at pH 2.0

The block-type chitosan was dissolved in McIlvaine buffer (pH 2.0) at 25 °C for 24 h (10 mg/mL). The chitosan sample (2 mg/mL) was incubated with Chia (50 μU) in a total volume of 50 μL and incubated at 37 °C for 1, 24, or 72 h.

### 4.6. Degradation of Random-Type Chitosan by Chia under Physiological pH Conditions

The random-type chitosan was dissolved in water at 25 °C for 24 h (10 mg/mL). The substrate solution (2 mg/mL) was incubated with Chia (50 μU) in 50 μL of McIlvaine buffer (pH 2.0, 5.0, or 7.0) at 37 °C for 1, 24, or 72 h.

### 4.7. Analysis of Chitooligosaccharides by FACE

The chitin and chitosan degradation products were analyzed by FACE as described initially by Jackson [54] and recently improved by our group [30,55,58]. This method can separate and detect very low amounts (~pmol) of chitooligosaccharides according to their molecular weight, based on their differential migration rates through polyacrylamide gel [30,54,55]. The samples were quantified using the Luminescent Image Analyzer (ImageQuant LAS 4000, GE Healthcare, Piscataway, NJ, USA) as described previously [30,32,55].

### 4.8. Statistical Analysis

We used the Student’s *t*-test for comparison between two sample groups. We considered the difference between comparisons to be significant when *p* < 0.05 for the statistical analyses.

## 5. Conclusions

Our study showed that Chia could degrade various chitosan substrates under physiological conditions corresponding to different mammalian tissues, including stomach and lung (pH 2–7). We also found that Chia is more efficient toward random-rather than block-type chitosan, a water-soluble molecule with potential for a wide range of applications, including agriculture, food, and biomedicine fields.

## Figures and Tables

**Figure 1 molecules-26-06706-f001:**
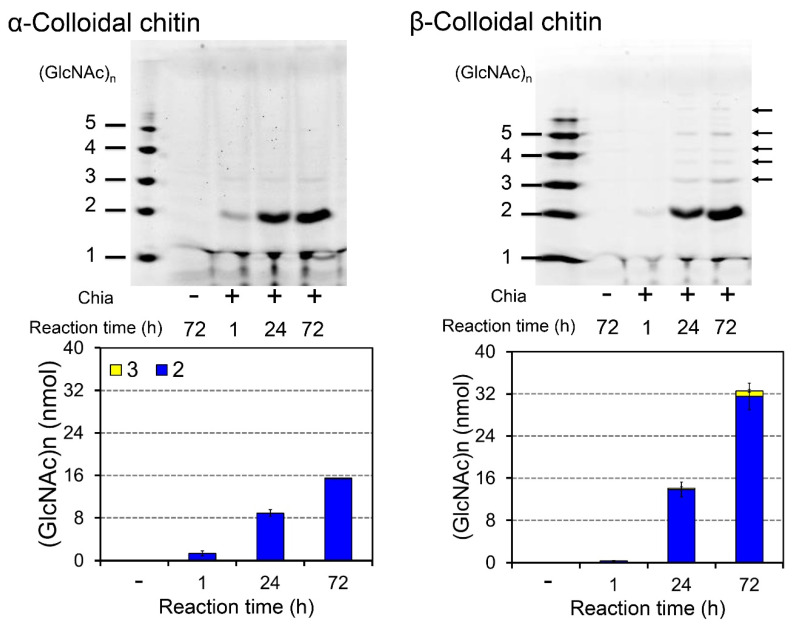
Degradation of α- and β-chitin substrates by Chia. α-colloidal or β-colloidal chitin was incubated with Chia at pH 2.0 for 1, 24, or 72 h. The resulting products were analyzed by the FACE method as described in the Materials and Methods. Chitin oligomers are shown in the left margin as standards. Recombinant Chia degraded α-chitin (**left**) and generated primarily dimers, whereas GlcNAc oligomers of variable sizes were produced from β-chitin (**right**). The data quatification is shown in lower panels.

**Figure 2 molecules-26-06706-f002:**
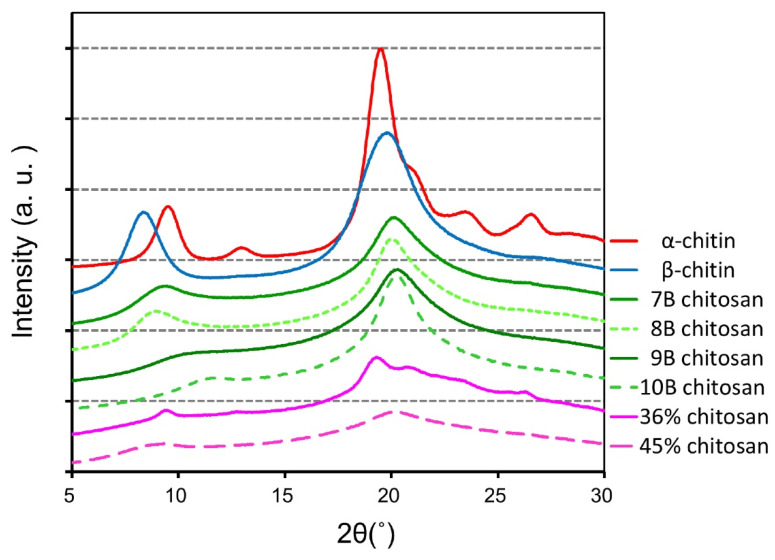
Observed X-ray diffraction patterns of block-type and random-type chitosan as well as α-and β-chitin substrates. The phase compositions of chitin and chitosan substrates were recorded at room temperature by X-ray diffraction.

**Figure 3 molecules-26-06706-f003:**
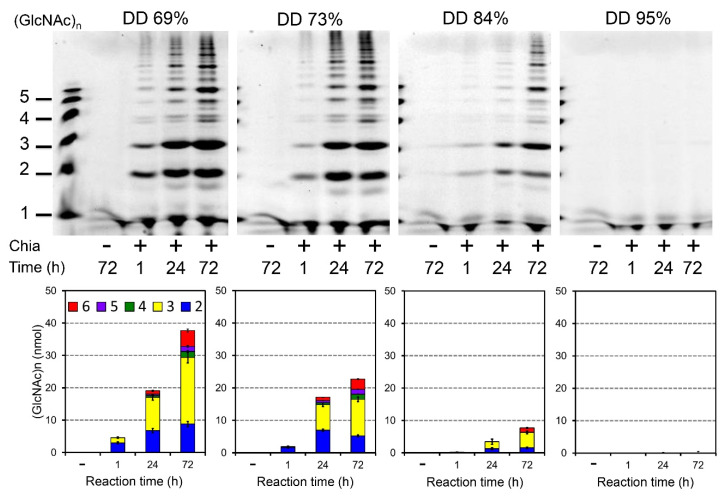
Degradation of block-type chitosan by Chia. Chitosan substrates (DD 69%, 73%, 84%, and 95%) were dissolved in McIlvaine buffer (pH 2.0) and incubated with mouse Chia at 37 °C for 1, 24, or 72 h and analyzed by the FACE method as described in the Materials and Methods. The degradation products obtained from block-type chitosan primarily consisted of (GlcNAc)_2_ to (GlcNAc)_9_ as well as longer chitooligosaccharides. Chia can degrade block-type chitosan with up to DD 84%, whereas it could not degrade DD 95%. Chitin oligomers are shown in the left margin as standards. Quantification of the data is shown in lower panels.

**Figure 4 molecules-26-06706-f004:**
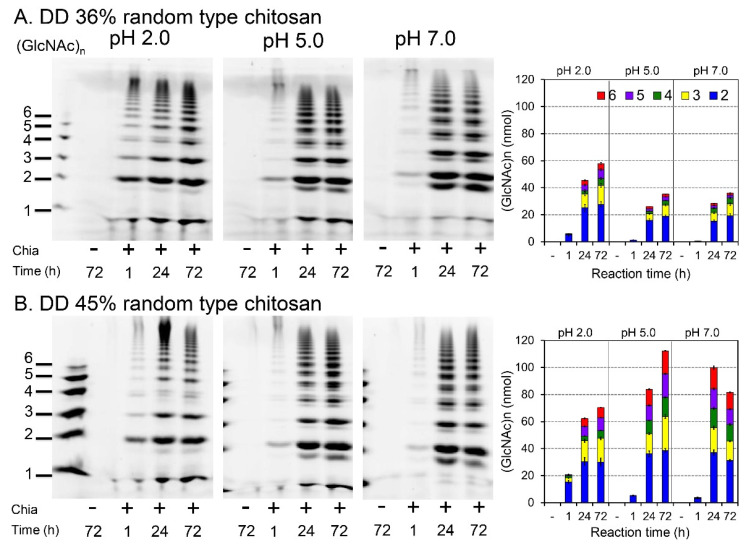
Degradation of random-type chitosan by Chia under different pH conditions. Chitosan was directly dissolved in water and incubated with Chia at pH 2.0, 5.0, or 7.0 and 37 °C for 1, 24, or 72 h, followed by the FACE method described in the Materials and Methods. Chitosan with DD 36% (**A**) and DD 45% (**B**) was used. Chitin oligomers are shown in the left margin as standards. Quantification of the data is shown in the right panels.

**Figure 5 molecules-26-06706-f005:**
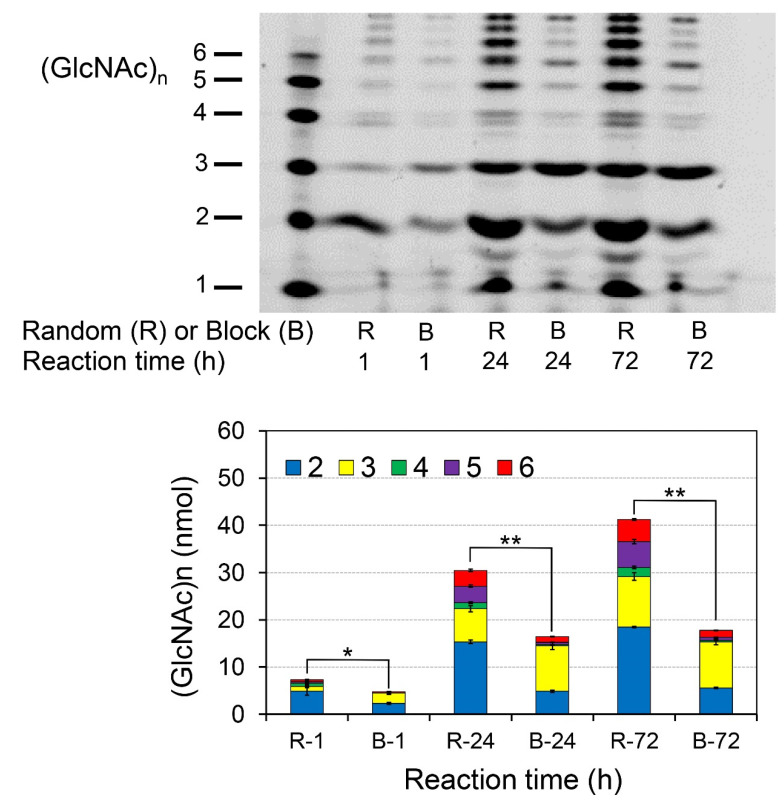
Chia produces chitooligosaccharides with more variable sizes from random-type chitosan when compared to the block-type form. Direct comparison of the degradation products from DD 69% block-type (B) and DD 45% random-type (R) chitosan substrates is essentially as described in Figure 3 and Figure 4. * *p* < 0.05; ** *p* < 0.01.

**Figure 6 molecules-26-06706-f006:**
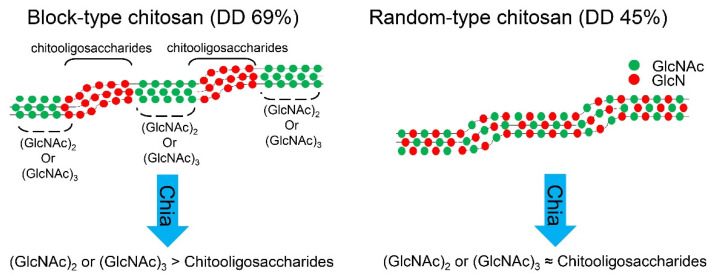
Chitosan structure and mode of degradation by Chia. The chitosan structure and mode of degradation by Chia are illustrated.

**Table 1 molecules-26-06706-t001:** Property of chitin and chitosan.

	DD (%)	Colloidal	Acidic Solution	Water
α-chitin	2.1	+	−	−
β-chitin	10.2	+	−	−
Block-type chitosan	69	−	+	−
	73	−	+	−
	84	−	+	−
	95	−	+	−
Random−type chitosan	35	−	+	+
	45	−	+	+

## Data Availability

Data supporting the reported results will be available with the corresponding author (Fumitaka Oyama).

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
