# Peer review of "Mouse Acidic Chitinase Effectively Degrades Random-Type Chitosan to Chitooligosaccharides of Variable Lengths under Stomach and Lung Tissue pH Conditions"

_molecules, 2021, doi:10.3390/molecules26216706_

Round 1
Reviewer 1 Report
In this manuscript of "Efficient production of variable length chitooligosaccharides from random-type chitosan using mouse acidic chitinase" authors prepared heterogeneously (block-type) and homogenously (random-type) deacetylated molecules and declared that mouse Chia digested chitosan into various chitooligosaccharides depending on the mode and degree of its deacetylation. As a whole, the manuscript lacks of novelty, the experimental design is relatively single and the content and results are not rich enough. Thus, it is suggest to reject the manuscript since the draft cannot reach the level of molecules.
Author Response
Point-by-Point Replies to Reviewers and Editor
Our responses to Reviewers and Editor in this letter and the revised manuscript are written in corresponding colors.
Brown: Reviewer #1; Red: Reviewer #2; Green: Reviewer #3; Blue: Editor
Reviewer #1
Comment 1:<In this manuscript of "Efficient production of variable length chitooligosaccharides from random-type chitosan using mouse acidic chitinase" authors prepared heterogeneously (block-type) and homogenously (random-type) deacetylated molecules and declared that mouse Chia digested chitosan into various chitooligosaccharides depending on the mode and degree of its deacetylation. As a whole, the manuscript lacks of novelty, the experimental design is relatively single and the content and results are not rich enough. Thus, it is suggest to reject the manuscript since the draft cannot reach the level of molecules.>
Response to Comment 1: Thank you for your comments. We further improved our original version based on all Reviewers’ comments and suggestions to show more profoundly the indicate novelty of our study in title, abstract and discussion in the revised version.
Reviewer 2 Report
In current study Wakita et al demonstrates that acidic chitinase produces chitooligosaccharides with higher efficieny and size variability from random-type chitosan when compared with the block-type form of the chitozan and propose explanations for observed experimental results.
Minor remarks:
- Generally manuscript is well written, however the text should be checked for grammatical and stylistic errors.
- The introduction and discussion section should be expanded with description of perspectives and practical implications of the findings. The limitations of current study should be added to the discussion of the article.
Author Response
Point-by-Point Replies to Reviewers and Editor
Our responses to Reviewers and Editor in this letter and the revised manuscript are written in corresponding colors.
Brown: Reviewer #1; Red: Reviewer #2; Green: Reviewer #3; Blue: Editor
Reviewer #2
Comment 1: <In current study Wakita et al demonstrates that acidic chitinase produces chitooligosaccharides with higher efficieny and size variability from random-type chitosan when compared with the block-type form of the chitozan and propose explanations for observed experimental results. Generally manuscript is well written, however the text should be checked for grammatical and stylistic errors.>
Response to Comment 1: Thank you for your comments and suggestions. We have checked grammatical and stylistic errors and corrected them in the revised version.
Comment 2: <The introduction and discussion section should be expanded with description of perspectives and practical implications of the findings. The limitations of current study should be added to the discussion of the article.>
Response to Comment 2: Thank you for your suggestions. We have improved the introduction and discussion to stress the expanded with the description of perspectives and practical implications of our findings in the introduction and discussion section of the revised version. We also included the limitations of the current study in the discussion in the revised version.
Reviewer 3 Report
Please mention clinical significance of this study.
Minor suggestion is to mention the rationale of study more clearly in abstract and conclusion.
Author Response
Point-by-Point Replies to Reviewers and Editor
Our responses to Reviewers and Editor in this letter and the revised manuscript are written in corresponding colors.
Brown: Reviewer #1; Red: Reviewer #2; Green: Reviewer #3; Blue: Editor
Reviewer #3
Comment 1: <Moderate English changes required>
Response to Comment 1: We carried out extensive English revisions in the revised version.
Comment 2: < Please mention clinical significance of this study.>
Response to Comment 2: Thank you for your suggestions. We have included the clinical significance of this study in the discussion section in the revised version.
Comment 3: <Minor suggestion is to mention the rationale of study more clearly in abstract and conclusion.>
Response to Comment 3: We have included the rationale of the study in the abstract and conclusion in the revised version.
Round 2
Reviewer 1 Report
Accept in present form